# Comparison of Two Nuclear Magnetic Resonance Spectroscopy Methods for the Measurement of Lipoprotein Particle Concentrations

**DOI:** 10.3390/biomedicines10071766

**Published:** 2022-07-21

**Authors:** Martin Rief, Reinhard Raggam, Peter Rief, Philipp Metnitz, Tatjana Stojakovic, Markus Reinthaler, Marianne Brodmann, Winfried März, Hubert Scharnagl, Günther Silbernagel

**Affiliations:** 1Division of General Anaesthesiology, Emergency- and Intensive Care Medicine, Department of Anaesthesiology and Intensive Care Medicine, Medical University of Graz, A-8036 Graz, Austria; martin.rief@medunigraz.at (M.R.); philipp.metnitz@medunigraz.at (P.M.); 2Division of Angiology, Department of Internal Medicine, Medical University of Graz, A-8036 Graz, Austria; reinhard.raggam@medunigraz.at (R.R.); peter.rief@medunigraz.at (P.R.); marianne.brodmann@medunigraz.at (M.B.); guenther.silbernagel@medunigraz.at (G.S.); 3Clinical Institute of Medical and Chemical Laboratory Diagnostics, University Hospital Graz, A-8036 Graz, Austria; tatjana.stojakovic@uniklinikum.kages.at; 4Department of Cardiology (CBF), Charité-Universitätsmedizin Berlin, 12203 Berlin, Germany; markus.reinthaler@charite.de; 5Institute of Biomaterial Science, Helmholtz-Zentrum Geesthacht, 14513 Teltow, Germany; 6Clinical Institute of Medical and Chemical Laboratory Diagnostics, Medical University of Graz, A-8036 Graz, Austria; winfried.maerz@medunigraz.at

**Keywords:** nuclear magnetic resonance spectroscopy, ultracentrifugation, lipoproteins, lipoprotein subclasses, methods, analysis, very-low-density lipoproteins, low-density lipoproteins, high-density lipoproteins

## Abstract

**Background:** Measuring lipoprotein particle concentrations may help to improve cardiovascular risk stratification. Both the lipofit (Numares) and lipoprofile (LabCorp) NMR methods are widely used for the quantification of lipoprotein particle concentrations. **Objective:** The aim of the present work was to perform a method comparison between the lipofit and lipoprofile NMR methods. In addition, there was the objective to compare lipofit and lipoprofile measurements of standard lipids with clinical chemistry-based results. **Methods:** Total, LDL, and HDL cholesterol and triglycerides were measured with ß-quantification in serum samples from 150 individuals. NMR measurements of standard lipids and lipoprotein particle concentrations were performed by Numares and LabCorp. **Results:** For both NMR methods, differences of mean concentrations compared to ß-quantification-derived measurements were ≤5.5% for all standard lipids. There was a strong correlation between ß-quantification- and NMR-derived measurements of total and LDL cholesterol and triglycerides (all r > 0.93). For both, the lipofit (r = 0.81) and lipoprofile (r = 0.84) methods, correlation coefficients were lower for HDL cholesterol. There was a reasonable correlation between LDL and HDL lipoprotein particle concentrations measured with both NMR methods (both r > 0.9). However, mean concentrations of major and subclass lipoprotein particle concentrations were not as strong. **Conclusions:** The present analysis suggests that reliable measurement of standard lipids is possible with these two NMR methods. Harmonization efforts would be needed for better comparability of particle concentration data.

## 1. Introduction

Measurement of lipoprotein particle concentrations in addition to standard lipids may help to estimate cardiovascular risk [1,2,3,4,5,6,7]. Historically, lipoprotein particle concentrations were analyzed with methods based on ultracentrifugation [8]. Another established method to measure lipoprotein particle concentrations is gradient gel electrophoresis [9,10]. However, ultracentrifugation and gradient gel electrophoresis are time-consuming and expensive. Therefore, ultracentrifugation and gradient gel electrophoresis are predominantly used for scientific purposes and not common in clinical routine. Today, several nuclear magnetic resonance (NMR)-based methods are available to measure lipoprotein particle concentrations [11,12]. The two most widespread methods for NMR-based quantification of lipoprotein particle concentrations have been introduced by LabCorp and Numares. There is broad evidence that high-density lipoprotein (HDL) particle concentrations measured with NMR may help to estimate cardiovascular risk [6,13]. Less consistently, low-density lipoprotein (LDL) particle concentrations have been suggested to be independently related to cardiovascular risk [5].

The comparability of different, advanced analytical methods for lipids and lipoproteins remains an issue [14,15]. So far, the comparability between different NMR methods has not been systematically investigated.

The aim of the present investigation was to provide an independent comparison of the two most widespread NMR-based methods for the quantification of lipoprotein particle concentrations. In addition to a comparison of the particle concentrations of the major lipoprotein particle classes, we aimed to compare NMR-based measurements of the major lipids (total cholesterol, LDL cholesterol, HDL cholesterol, and triglycerides) with standard clinical chemistry-based measurements (ß-quantification). Moreover, we aimed to analyze internal correlation matrices for the two NMR methods. Finally, we aimed to perform a comparison of corresponding lipoprotein subclass particle concentrations.

## 2. Methods

### 2.1. Study Participants

Recruitment of one hundred fifty study participants was performed at the outpatient clinic of the Division of Angiology (of the Department of Internal Medicine) and the preoperative anaesthesia outpatient clinic of the Division of General Anaesthesiology, Emergency and Intensive Care Medicine (of the Department of Anaesthesiology and Intensive Care Medicine) at the University Hospital Graz, Austria, between 17 February and 30 June 2020. The inclusion criteria were age ≥ 18 years. There were no exclusion criteria. Written informed consent was obtained from each participant. Total cholesterol ranged between 93 and 339 mg/dL.

### 2.2. Blood Sampling

We collected blood samples as a part of blood withdrawal in clinical routine. Overnight fasting was not required, as the samples were only used for a method comparison. In addition, fasting is not required according to the current recommendations of the European Atherosclerosis Society and the European Federation of Clinical Chemistry and Laboratory Medicine [16].

### 2.3. Specimen Material

From each patient, two tubes of 9 mL of whole blood were collected with Greiner bio-one^®^ Vacuette Z Serum (red, 9.0 mL; 455092). Serum was separated from blood cells by centrifugation (10 min at 6490 rpm and 15 °C) and aliquoted in cryotubes (Nunc Universal^®^ 1.8 mL).

### 2.4. Storage of Samples and Shipping

The cryotubes were cooled down and stored at −80 °C. The 150 samples of 1 mL serum each were sent in accordance with international security regulations for medical specimens to the laboratories in frozen condition.

### 2.5. Laboratory Measurements

In this study, three different laboratories performed the analyses. Standard lipid measurements were performed at the Clinical Institute of Medical and Chemical Laboratory Diagnostics of the Medical University of Graz, Austria. Lipoproteins were separated using a combined ultracentrifugation–precipitation method (ß-quantification). The VLDL fraction (d < 1.006 g/mL) was removed after ultracentrifugation (18 h, 10 °C, 98,000× *g*). ApoB-containing lipoproteins in the resulting bottom fraction were precipitated using phosphotungstic acid with the HDL particles remaining in solution. LDLC was calculated by subtracting cholesterol after precipitation from the respective concentrations before precipitation. Cholesterol and triglycerides were measured with enzymatic reagents from Diasys (Holzheim, Germany) on an Olympus AU680 analyzer [8]. The LabCorp Corp. (100 Perimeter Park, Morrisville, 27560 North Carolina, USA) performed NMR analysis using the NMR LipoProfile^®^ LP4 method (in the further text briefly lipoprofile) in Raleigh [11]. Particle concentrations of lipoproteins of different sizes were calculated from the measured amplitudes of their spectroscopically distinct lipid methyl group NMR signals. Weighted-average lipoprotein particle sizes are derived from the sum of the diameter of each subclass multiplied by its relative mass percentage based on the amplitude of its methyl NMR signal [4]. The Numares AG (Am Biopark 9, 93053 Regensburg, Germany) performed NMR analysis using the AXINON^®^ lipoFIT^®^ method (in the further text briefly lipofit) in Regensburg with an Avance III HD nuclear magnetic resonance spectrometer (Bruker; Billerica, MA, USA), an Ascend 600 MHz magnet (Bruker), and using TopSpin 3.2 (Bruker) and Axinon Suite 1.0.0.1 (Numares, Regensburg, Germany) software [12,17].

### 2.6. Statistical Methods

LabCorp provided duplicate measurements, of which the first measurements were used for analyses. The measurements are given as means and standard deviations. Associations among clinical chemistry and NMR-based standard lipid measurements are given as Pearson and Spearman correlations. Likewise, comparisons between the Numares (lipofit) and LabCorp (lipoprofile) measurements were analyzed using Pearson correlation coefficients and non-parametric Passing–Bablok regression. The analysis plan has been pre-specified. The statistical package from IBM^®^ (IBM Corp. Released 2019. IBM SPSS Statistics for Windows, Version 26.0. Armonk, NY, USA: IBM Corp) was used. Passing–Bablok regression was calculated with the Analyse-it Method Validation Edition for Microsoft Excel 5.90 (Analyse-it Software Ltd., Leeds, UK)

### 2.7. Ethical and Regulatory Aspects

The study was approved by the ethics committee of the Medical University of Graz (29-479 ex 16/17) and the Federal Office for Safety in Health Care of Austria (Bundesministerium für Sicherheit im Gesundheitswesen—BASG); Agency for Health and Food Security (AGES) on 15 May 2019 (ref. No. 11458092). The study was performed in accordance with the Declaration of Helsinki, and all participants gave written, informed consent.

## 3. Results

### 3.1. Raw Data

The entire anonymized raw data file is provided in the online supplements. All measurements were complete with the lipoprofile NMR method. In two samples, the values for small LDL particles, and in three samples, the values for small HDL particles were missing with the lipofit method. For several parameters, few values were below the detection limit with the Numares NMR (lipofit) method (Appendix A).

### 3.2. Standard Lipid Measured with ß-Quantification, Lipofit NMR and Lipoprofile NMR

For both NMR methods, differences of mean concentrations compared to ß-quantification-derived measurements were ≤5.5% for all standard lipids. Total cholesterol was lower with both NMR methods compared with the enzymatic assay. Triglycerides were modestly higher with both NMR methods compared with the enzymatic assay (Table 1). Correlations between the standard method and both the lipoprofile and lipofit methods were strong for total cholesterol, LDL cholesterol and triglycerides with marginally higher correlation coefficients for the lipoprofile method. Correlations with ß-quantification-derived HDL cholesterol were weaker for both the lipofit and the lipoprofile methods (Figure 1a,b, Appendix A). It can also be seen from Figure 1b that there are few downward outliers with the lipoprofile NMR triglyceride measurement. Correlations of standard lipids measurements between the lipoprofile and lipofit methods were strong (Appendix A).

In two samples, HDL cholesterol measured with the enzymatic method was considerably higher compared with the two NMR methods (Appendix A, Figure 1b).

### 3.3. Correlations among Lipids and Lipoprotein Particles within the Lipofit NMR Method and within the Lipoprofile NMR Method

Overall, internal correlation matrices appeared consistent when comparing the lipoprofile and lipofit methods (Table 2 and Table 3). With the lipofit method, LDL particles were positively associated with total cholesterol and triglycerides. They were also positively associated with large VLDL particles and HDL particles. Large LDL particles were only weakly associated with small LDL particles and positively associated with HDL particles. Small LDL particles were inversely related to LDL size. HDL particles were not associated with triglycerides. Large VLDL particles were inversely related to HDL cholesterol. LDL size was positively associated with HDL size (Table 2). With the lipoprofile method, LDL particles were also positively associated with total cholesterol and triglycerides. They were modestly and positively associated with large VLDL particles and HDL particles. Large LDL particles were not significantly associated with small LDL particles and HDL particles. Small LDL particles were inversely related to LDL size. HDL particles were also not associated with triglycerides. Large VLDL particles were also inversely related to HDL cholesterol. LDL size was also positively associated with HDL size (Table 3 and Appendix A).

### 3.4. Lipoprotein Particles Measured with Lipofit NMR and Lipoprofile NMR

The correlations of the LDL and HDL lipoprotein particle concentrations were strong between the lipofit and lipoprofile methods (Appendix A). The mean LDL particle concentration was 13% higher with the lipoprofile method, whereas the HDL particle concentration was markedly higher with the lipofit method (+55%). Total VLDL particles could not be compared as this parameter is not available for the lipofit method. There was good agreement between mean LDL size measured with the lipofit and lipoprofile methods (Table 1). Correlations between the two NMR methods were moderate for LDL size but stronger for HDL size (Appendix A).

Based on size categorization, small and medium lipoprofile LDL particles were compared with small lipofit LDL particles (Appendix A). With the lipofit method, the mean concentration of large LDL particles was higher than the mean concentration of small LDL particles. With the lipoprofile method, on the other hand, small LDL particles were the predominant subclass (Table 1). The correlations of the LDL subclass particle concentrations between the two NMR methods were only moderate, especially for large LDL particles (Appendix A, Figure 2 and Figure 3). We have performed further calculations with alternative size categories, which did not correspond to the predefined study protocol. Combining medium and large LDL particles with the lipoprofile method to compare them with large lipofit LDL particles resulted in mean values of 909 (±442) nmol/L for the lipoprofile method and 677 (±289) nmol/L for the lipofit method and a correlation of 0.771. Comparing small lipoprofile to small lipofit LDL particles, there was better agreement for mean concentrations with 528 (±270) nmol/L for lipoprofile and 421 (±266) nmol/L for lipofit. However, the correlation coefficient was only 0.555.

Consistent with the lower total HDL particle concentration, the concentrations of HDL subclass particles were also substantially lower with the lipoprofile compared to lipofit method (Table 1). The correlations were stronger for large HDL particles than for small HDL particles (Appendix A, Figure 2 and Figure 3).

The concentration of large VLDL particles was lower with the lipoprofile method compared with the lipofit method, with weak correlations between the two methods (Table 1).

## 4. Discussion

This is the first independent and systematic comparison of the two most widespread NMR methods for the quantification of lipoprotein particle concentrations. Both the lipofit and the lipoprofile methods showed strong correlations with ß-quantification for total, LDL, and HDL cholesterol, and for triglycerides. A few downward outliers for triglycerides measured with the lipoprofile NMR method may be due to the effects of freezing on triglyceride-rich particles [11]. The relatively high difference in HDL cholesterol values between the enzymatic method and both NMR methods in some samples may be due to limitations of the precipitation step. The differences of mean concentrations compared to ß-quantification-derived measurements were ≤5.5% for all standard lipids, both for the lipofit and the lipoprofile methods. Hence, the two NMR methods appear to provide reliable information on the concentration of standard lipids.

The main objective of the present study was to compare the results of the two NMR methods with regard to lipoprotein particle concentrations. In fact, there were acceptable correlations of the LDL and HDL particle concentrations between the two NMR methods. Whereas LDL particle concentrations were similar, there were differences for the mean concentrations of HDL particles with the lipofit method reporting higher values.

Regarding lipoprotein subclass particle concentrations, it has to be considered that the lipofit and lipoprofile categorizations differ considerably. Hence, direct comparisons were not feasible. Rather, we aimed to compare roughly corresponding, partly combined lipoprotein subclass categories. Relatively weak concordance between the two NMR methods was particularly observed for small and large LDL particle concentrations, with the lipoprofile NMR method showing a higher proportion of small LDL particles and the lipofit NMR method showing a higher proportion of large LDL particles. This may be due to differences in classifications, since the results were more consistent when medium and large LDL particles of the lipoprofile method were compared with large lipofit LDL particles instead of combining medium with small lipoprofile LDL particles. As observed for total HDL particles, small and large HDL particle concentrations were higher with the lipofit method. These differences may also be due to calibration but cannot be definitely explained.

The concentration of total VLDL particles is not provided by the lipofit method so that only large VLDL particles could be compared. Although there appeared to be a strong correlation, the mean particle concentration of large VLDL particles was higher for the lipofit method.

High concordance between the two NMR methods was observed for the mean values of the LDL and HDL sizes. However, the correlations were only moderate.

Comparing the lipofit and lipoprofile analyte panel, lipoprofile provides a more comprehensive list of parameters. It includes a more detailed separation of lipoprotein subclasses and also provides information on apolipoprotein B.

The internal correlation matrix among major lipids and lipoprotein particles gave similar results for the lipofit and the lipoprofile methods. This supports that measurement of lipoprotein particles with these methods is comparable. Most importantly, there should not be a strong positive correlation between large LDL particles and small LDL particles [18]. No such correlation was seen with the lipoprofile and lipofit methods. This is in contrast to a recent analysis with the Nightingale method, which showed strong, positive correlations among all LDL subclasses (r > 0.8) [19]. Still, the differences in mean particle concentrations between the lipofit and lipoprofile methods require further investigations. This is of particular relevance, as a more precise characterization of the lipoprotein profile with NMR may help to improve risk classification reflecting different pathophysiological features of the various lipoprotein subclasses [20]. This may also be of relevance in times when an increasing number of drugs is available to treat distinct lipid disorders, e.g., therapies addressing apolipoprotein C-III for familial hyperchylomicronaemia [21]. It is also an advantage that the NMR methods are not time-consuming and a large number of samples can be analyzed in a relatively short period of time. This makes them useful for scientific purposes, considering that standard procedures to analyze lipoprotein metabolism such as analytical ultracentrifugation are very time-consuming and expensive. It is a disadvantage of the NMR methods that they are not routinely available in standard laboratories because they require special equipment. Moreover, a large sample size is necessary (~0.5–1 mg that is dissolved in ~0.5 mL of solvent) for the analysis. The lack of harmonization between the different providers also makes it difficult to interpret and compare certain results. Moreover, the associations of certain lipoprotein particle concentrations with cardiovascular endpoints have been inconsistent [1,2,3,4,5,6,7,8]. Therefore, particle concentrations from NMR measurements are currently also not recommended as therapeutic targets in the guidelines for the treatment of dyslipidaemia endorsed by the European Atherosclerosis Society and the European Society of Cardiology [21].

Regarding the harmonization of NMR methods, using the same particle size cutoff values could be a first step. In addition, further work would have to clarify whether the harmonization of the cutoff values yields different results or whether and to what extent the mathematical algorithm underlying the different NMR methods may be the primary cause of the deviating results. However, a harmonization effort would necessarily be voluntary until some sort of regulatory framework is imposed, for example, by the Food and Drug Administration. So near-term prospects are not great. Nevertheless, the need for such regulation, which appears years away, is at least recognized by the IFCC Metabolomics Working Group.

The strong aspect of this study is that we performed an independent method comparison of the newest versions of the two most widespread NMR methods for the quantification of lipoprotein particle concentrations.

One limitation of the study is that the ß-quantification method provides results for the main lipoprotein fractions (VLDL, LDL, HDL) only. Hence, we were not able to provide an independent method comparison for the analysis of LDL and HDL subclass particles. However, the primary focus was to address the comparability of the two widespread NMR methods.

## 5. Conclusions

To sum up, the present study shows that standard lipids can be reliably measured with NMR methods. Harmonization efforts for better comparability of lipoprotein particle concentrations measured with NMR are encouraged.

## Figures and Tables

**Figure 1 biomedicines-10-01766-f001:**
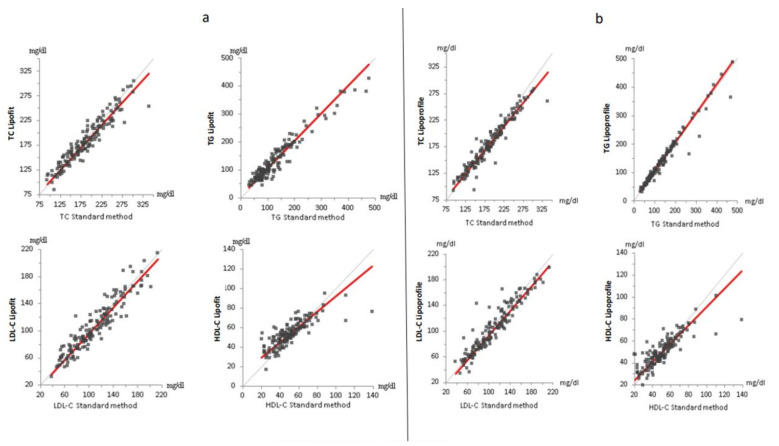
Comparison of standard lipids between ß-quantification and lipofit NMR (**a**) and lipoprofile NMR (**b**). The (**a**,**b**) show the Passing–Bablok regression for total cholesterol (top left), triglycerides (top right), low-density lipoprotein cholesterol (bottom left), and high-density lipoprotein cholesterol (bottom right). The respective slopes of the regression lines (red) were 0.889 (TC), 1.027 (TG), 0.960 (LDL-C), and 0.830 (HDL-C) for lipofit and 0.914 (TC), 0.983 (TG), 0.973 (LDL-C), and 0.786 (HDL-C) for lipoprofile, respectively. The grey line represents the line of identity. (C = cholesterol. HDL = high-density lipoproteins. LDL = low-density lipoproteins. TC = total cholesterol. TG = triglycerides).

**Figure 2 biomedicines-10-01766-f002:**
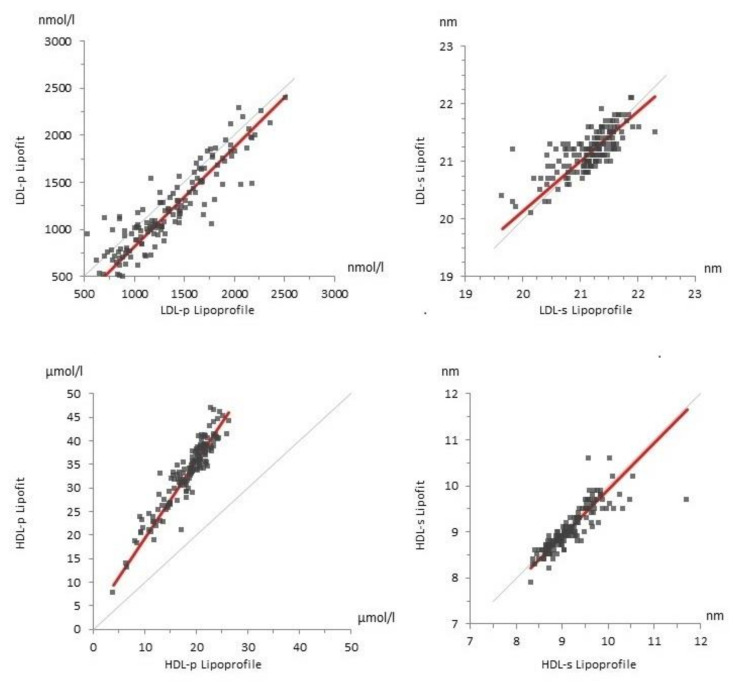
Comparison of lipoprotein particles between the lipoprofile NMR and lipofit NMR methods. The figures show the Passing–Bablok regression for LDL-p (top **left**), LDL-s (top right), HDL-p (bottom left, and HDL-s (bottom **right**). The respective slopes of the regression lines (red) were 1.057 (LDL-p), 0.860 (LDL-s), 1.637 (HDL-p), and 1.014 (HDL-s), respectively. The grey line represents the line of identity. (HDL = high-density lipoproteins. LDL = low-density lipoproteins. p = particles. s = size).

**Figure 3 biomedicines-10-01766-f003:**
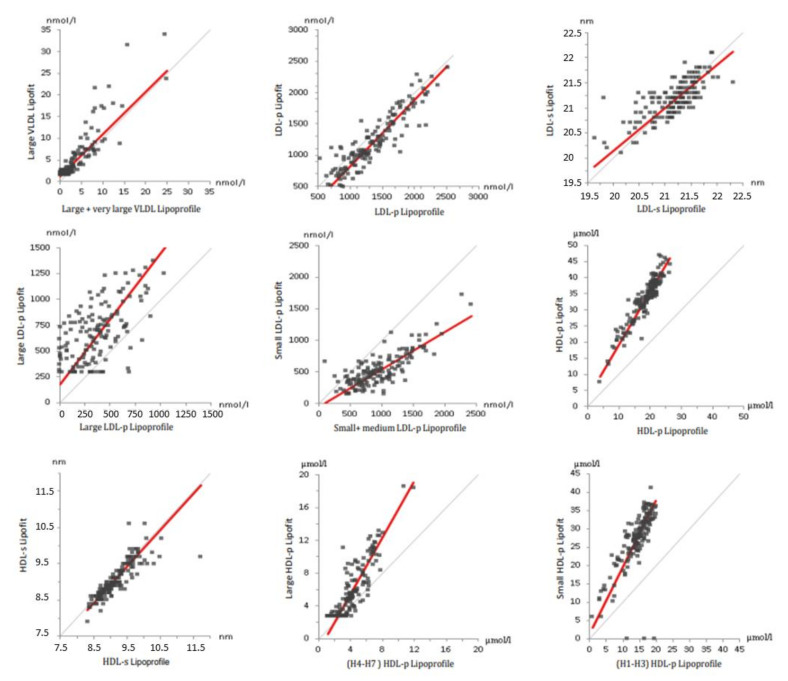
Comparison of lipoprotein particles between the lipoprofile and lipofit methods. The figures show the Passing–Bablok regression for large and very large VLDL (top **left**), LDL-p (top middle), LDL-s (top **right**), large LDL-p (middle **left**), small and medium LDL-p (**middle**), HDL-p (middle right), HDL-s (bottom **left**), large HDL-p (bottom middle), and small HDL-p (bottom **right**). The respective slopes of the regression lines (red) were 0.980 (large VLDL-p), 1.272 (large LDL-p), 0.593 (small and medium LDL-p), 1.817 (small HDL-p) and 1.722 (large HDL-p), respectively. The grey line represents the line of identity. (HDL = high-density lipoproteins. LDL = low-density lipoproteins. p = particles. VLDL = very-low-density lipoproteins).

**Table 1 biomedicines-10-01766-t001:** Mean concentrations of standard lipids and lipoprotein (sub)classes.

Parameter	Units	Lipoprofile	Lipofit	Standard Method
**TC**	mg/dL	180 (±45)	183 (±46)	188 (±51)
**TG**	mg/dL	135 (±81)	135 (±78)	132 (±84)
**LDL-C**	mg/dL	105 (±37)	107 (±38)	110 (±38)
**HDL-C**	mg/dL	50.1 (±13.2)	53.5 (±12.7)	50.7 (±17.7)
**LVLDL-p**	nmol/L	3.42 (±4.18) ^a^	4.94 (±5.6)	
**LDL-p**	nmol/L	1330 (±444)	1176 (±465)	
**LDL-size**	nm	21,1 (±0,56)	21.2 (±0.37)	
**LLDL-p**	nmol/L	377 (±241)	677 (±289)	
**SLDL-p**	nmol/L	953 (±407) ^b^	528 (±270)	
**HDL-p**	µmol/L	18.4 (±4.4)	33.2 (±7.1)	
**HDL-size**	nm	9.19 (±0.51)	9.10 (±0.47)	
**LHDL-p**	µmol/L	4.37 (±1.87) ^c^	6.08 (±3.31)	
**SHDL-p**	µmol/L	14.0 (±4.0) ^d^	26.8 (±7.6)	

(^a^ large + very large VLDL-p. ^b^ small + medium LDL-p. ^c^ H4-H7 HDL-p. ^d^ H1-H3 HDL-p. C = cholesterol. HDL = high-density lipoproteins. LDL = low-density lipoproteins. LHDL = large high-density lipoproteins. LLDL = large low-density lipoproteins. LVLDL = large very-low-density lipoproteins. p = particles. s = size. SHDL = small high-density lipoproteins. SLDL = small low-density lipoproteins. TC = total cholesterol. TG = triglycerides).

**Table 2 biomedicines-10-01766-t002:** Internal correlation matrix among lipids and lipoprotein particles for the lipofit method.

Lipofit	LVLDL-p	p	LDL-p	p	LLDL-p ^a^	p	SLDL-p ^a^	p	HDL-p	p	LHDL-p	p	SHDL-p ^b^	p	TC	p	TG	p	HDL-C	p	LDL-C	p	LDLsize	p	HDLsize	p
**LVLDL-p**	**1**		0.34	<0.001	−0.01	0.928	0.56	<0.001	0.07	0.365	−0.29	<0.001	0.23	0.005	0.15	0.07	0.9	<0.001	−0.28	<0.001	0.05	0.582	−0.47	<0.001	−0.34	<0.001
**LDL-p**	0.34	<0.001	**1**		0.8	<0.001	0.74	<0.001	0.32	<0.001	−0.15	0.069	0.42	<0.001	0.87	<0.001	0.54	<0.001	0.06	0.485	0.91	<0.001	−0.11	0.188	−0.34	<0.001
**LLDL-p ^a^**	−0.01	0.928	0.8	<0.001	**1**		0.21	<0.011	0.42	<0.001	0.26	<0.002	0.31	<0.001	0.85	<0.001	0.18	0.029	0.41	<0.001	0.83	<0.001	0.35	<0.001	0.04	0.634
**SLDL-p ^a^**	0.56	<0.001	0.74	<0.001	0.21	0.011	**1**		0.02	0.859	−0.61	<0.001	0.36	<0.001	0.49	<0.001	0.69	<0.001	−0.4	<0.001	0.56	<0.001	−0.56	<0.001	−0.69	<0.001
**HDL-p**	0.07	0.365	0.32	<0.001	0.42	<0.001	0.02	0.859	**1**		0.3	<0.001	0.86	<0.001	0.49	<0.001	0.11	0.17	0.7	<0.001	0.32	<0.001	0.04	0.661	−0.01	0.877
**LHDL-p**	−0.29	<0.001	−0.15	0.069	0.26	0.002	−0.61	<0.001	0.3	<0.001	**1**		−0.22	0.009	0.15	0.069	−0.38	<0.001	0.81	<0.001	−0.02	0.841	0.51	<0.001	0.92	<0.001
**SHDL-p ^b^**	0.23	0.005	0.42	<0.001	0.31	<0.001	0.36	<0.001	0.86	<0.001	−0.22	0.009	**1**		0.42	<0.001	0.33	<0.001	0.29	<0.001	0.35	<0.001	−0.23	0.005	−0.52	<0.001
**TC**	0.15	0.07	0.87	<0.001	0.85	<0.001	0.49	<0.001	0.49	<0.001	0.15	0.069	0.42	<0.001	**1**		0.36	<0.001	0.44	<0.001	0.96	<0.001	0.25	0.003	−0.03	0.717
**TG**	0.9	<0.001	0.54	<0.001	0.18	0.029	0.69	<0.001	0.11	0.17	−0.38	<0.001	0.33	<0.001	0.36	<0.001	**1**		−0.29	<0.001	0.28	<0.001	−0.42	<0.001	−0.5	<0.001
**HDL-C**	−0.28	<0.001	0.06	0.485	0.41	<0.001	−0.4	<0.001	0.7	<0.001	0.81	<0.001	0.29	<0.001	0.44	<0.001	−0.29	<0.001	**1**		0.25	0.002	0.51	<0.001	0.64	<0.001
**LDL-C**	0.05	0.582	0.91	<0.001	0.83	<0.001	0.56	<0.001	0.32	<0.001	−0.2	0.841	0.35	<0.001	0.96	<0.001	0.28	<0.001	0.25	0.002	**1**		0.19	0.021	−0.16	0.053
**LDL size**	−0.47	<0.001	−0.11	0.188	0.35	<0.001	−0.56	<0.001	0.04	0.661	0.51	<0.001	−0.23	0.005	0.25	0.003	−0.42	<0.001	0.51	<0.001	0.19	0.021	**1**		0.56	<0.001
**HDL size**	−0.36	0.1	−0.34	<0.001	0.04	0.634	−0.69	<0.001	−0.01	0.877	0.92	<0.001	−0.52	<0.001	−0.03	0.717	−0.52	<0.001	0.65	<0.001	−0.16	0.034	0.56	<0.001	**1**	

(^a^ 148 values. ^b^ 147 values. C = cholesterol. HDL = high-density lipoproteins. LDL = low-density lipoproteins. LHDL = large high-density lipoproteins. LLDL = large low-density lipoproteins. LVLDL = large very-low-density lipoproteins. *p* = particles. SHDL = small high-density lipoproteins. SLDL = small low-density lipoproteins. TC = total cholesterol. TG = triglycerides).

**Table 3 biomedicines-10-01766-t003:** Internal correlation matrix among lipids and lipoprotein particles for the lipoprofile method.

Lipoprofile	LVLDL-p	p	LDL-p	p	LLDL-p	p	SLDL-p	p	HDL-p	p	LHDL-p	p	SHDL-p	p	TC	p	TG	p	HDL-C	p	LDL-C	p	LDL-s	p	HDL-s	p
**LVLDL-p**	**1**		0.18	0.031	−0.35	<0.001	0.4	<0.001	0.09	0.267	−0.35	<0.001	0.23	0.003	0.2	0.016	0.81	<0.001	−0.32	<0.001	0.06	0.451	−0.46	<0.001	−0.42	<0.001
**LDL-p**	0.18	0.031	**1**		0.42	<0.001	0.31	<0.001	0.3	<0.001	−0.12	0.14	0.3	<0.001	0.9	<0.001	0.35	<0.001	0.07	0.368	0.92	<0.001	0.08	0.345	−0.32	<0.001
**LLDL-p**	−0.35	<0.001	0.42	<0.001	**1**		−0.13	0.106	0.01	0.902	0.25	0.002	−0.11	0.198	0.52	<0.001	−0.28	0.001	0.32	<0.001	0.61	<0.001	0.79	<0.001	0.3	<0.001
**SLDL-p**	0.4	<0.001	0.31	<0.001	−0.13	0.11	**1**		0.33	<0.001	−0.26	0.001	0.48	<0.001	0.67	<0.001	0.54	<0.001	−0.11	0.198	0.64	<0.001	−0.67	<0.001	−0.29	<0.001
**HDL-p**	0.09	0.267	0.3	<0.001	0.01	0.902	0.33	<0.001	**1**		0.41	<0.001	0.91	<0.001	0.48	<0.001	0.07	0.375	0.64	<0.001	0.32	<0.001	0.05	0.527	−0.35	<0.001
**LHDL-p**	−0.35	<0.001	−0.12	0.14	0.25	0.002	−0.26	0.001	0.41	<0.001	**1**		−0.02	0.812	0.14	0.091	−0.31	<0.001	0.86	<0.001	−0.02	0.799	0.32	<0.001	0.73	<0.001
**SHDL-p**	0.23	0.003	0.3	<0.001	−0.11	0.198	0.48	<0.001	0.91	<0.001	−0.02	0.812	**1**		0.46	<0.001	0.22	0.006	0.3	<0.001	0.36	<0.001	−0.13	0.122	−0.6	<0.001
**TC**	0.2	0.016	0.9	<0.001	0.52	<0.001	0.67	<0.001	0.48	<0.001	0.14	0.091	0.46	<0.001	**1**		0.35	<0.001	0.34	<0.001	0.95	<0.001	0.25	0.002	−0.16	0.048
**TG**	0.81	<0.001	0.35	<0.001	−0.28	0.001	0.54	<0.001	0.07	0.375	−0.31	<0.001	0.22	0.006	0.35	<0.001	**1**		−0.35	<0.001	0.19	0.021	−0.5	<0.001	−0.46	<0.001
**HDL-C**	−0.32	<0.001	0.07	0.368	0.32	<0.001	−0.11	0.198	0.64	<0.001	0.86	<0.001	0.3	<0.001	0.34	<0.001	−0.35	<0.001	**1**		0.18	0.026	0.38	<0.001	0.4	<0.001
**LDL-C**	0.06	0.451	0.92	<0.001	0.61	<0.001	0.64	<0.001	0.32	<0.001	−0.02	0.799	0.36	<0.001	0.95	<0.001	0.19	0.021	0.18	0.026	**1**		0.34	<0.001	−0.17	0.039
**LDL-s**	−0.46	<0.001	0.08	0.345	0.79	<0.001	−0.67	<0.001	0.05	0.527	0.32	<0.001	−0.13	0.122	0.25	0.002	−0.5	<0.001	0.38	<0.001	0.34	<0.001	**1**		0.38	<0.001
**HDL-s**	−0.42	<0.001	−0.32	<0.001	0.3	<0.001	−0.29	<0.001	−0.35	<0.001	0.73	<0.001	−0.6	<0.001	−0.16	0.048	−0.46	<0.001	0.4	<0.001	−0.17	0.039	0.38	<0.001	**1**	

(C = cholesterol. HDL = high-density lipoproteins. LDL = low-density lipoproteins. LHDL = large high-density lipoproteins. LLDL = large low-density lipoproteins. LVLDL = large very-low-density lipoproteins. p = particles. SHDL = small high-density lipoproteins. SLDL = small low-density lipoproteins. TC = total cholesterol. TG = triglycerides).

## Data Availability

The data presented in this study are available on request from the corresponding author.

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
