# Peer review of "Comparison of Two Nuclear Magnetic Resonance Spectroscopy Methods for the Measurement of Lipoprotein Particle Concentrations"

_biomedicines, 2022, doi:10.3390/biomedicines10071766_

Round 1

Reviewer 1 Report

The authors of the current manuscript aimed to compare two NMR methods to determine lipoprotein particle concentration. The topic is interesting, but the manuscript has some major problems.

1.     The two methods performed by the two companies (Numares and LabCorp) are very briefly described and no major differences were highlighted in the Methods Chapter. This is very important since this is the basis of the entire paper. The two companies use two different kits. Thus, it would be proper to name the kits, not the companies. The readers would want to be informed about the performances of the kits, not about a measurement done by a specific company.

2.     Table 1. Which is the size of the LVLDL-p, LLDL-p, SLDL-p, LHDL-p, and SHDL-p? Maybe there is a distinct issue since the differences appear when lipoproteins large and small were determined (i.e. LLDL-p, SLDL-p, SHDL-p).

3.     Tables 2-4 may be replaced by the corresponding Supplemental Figures since those are more illustrative. Tables 2-4 should become Supplemental Tables. However, the Figure legends should be more descriptive (including Figure 1 legend).

4.     The conclusion of the study refers to the reliability of the NMR methods for the measurement of the standard lipids and not for the other lipoprotein particles analyzed. However, NMR is a method that needs complex equipment, and thus I cannot see the advantages to use it to determine the lipoproteins in serum. Detail the advantages and disadvantages and comment on how the harmonization of the two methodologies should be done.  

Minor concerns

1.     Why it is noted that overnight fasting is not required? (page 3 Blood sampling)

2.     Mention G instead of rpm for all the centrifugation steps.

3.     Mention which value is considered statistically significant in the Statistical methods for all the methods described.

Reviewer 2 Report

The authors presented a method comparison between two RMN methods (Numares vs LabCorp) for the measurements of lipids and lipoprotein concentrations as well as lipoprotein particles in concentration and size. The study is very well conducted in number of samples and parameter comparison. This study is a good approach to understand the clinical significance of Lipoprotein particles as risk factors for CVD.

Minor considerations:

1.- Table 4 and Table 5: the internal correlation analysis is very difficult to interpret in reference with the phrase: “…were consistent except for large LDL particles comparing….”. These results should be detailed to understand what is comparing and which is the data most significant to show.

2.- Figure 1: in the legend the LDL-s and HDL-s should be mentioned.

3.- Supp. Figure 3: It would be recommendable to include this supplementary figure as a Figure 2 and explain these results in the main text to facilitate the total comparison  between both methods.

Round 2

Reviewer 1 Report

The authors gave pertinent answers and the manuscript was revised, accordingly. The manuscript can be published in its current form.